# Long-Term Tumor-Targeting Effect of *E. coli* as a Drug Delivery System

**DOI:** 10.3390/ph17040421

**Published:** 2024-03-26

**Authors:** Gun Gyun Kim, Hongje Lee, Dan Bi Jeong, Sang Wook Kim, Jae-Seon So

**Affiliations:** 1Department of Nuclear Medicine, Dongnam Institute of Radiological and Medical Sciences, Busan 46033, Republic of Korea; ggkim88@dirams.re.kr (G.G.K.); hjlee@dirams.org (H.L.); danbee818@dirams.re.kr (D.B.J.); 2Department of Advanced Materials Chemistry, Dongguk University, Gyeongju 38066, Republic of Korea; 3Department of Medical Biotechnology, Dongguk University, Gyeongju 38066, Republic of Korea

**Keywords:** radiotracer, tumor targeting, zirconium-89, *Escherichia coli*, PET imaging, drug delivery

## Abstract

To overcome the limitations of current nano/micro-scale drug delivery systems, an *Escherichia coli* (*E. coli*)-based drug delivery system could be a potential alternative, and an effective tumor-targeting delivery system can be developed by attempting to perform chemical binding to the primary amine group of a cell membrane protein. In addition, positron emission tomography (PET) is a representative non-invasive imaging technology and is actively used in the field of drug delivery along with radioisotopes capable of long-term tracking, such as zirconium-89 (^89^Zr). The membrane proteins were labeled with ^89^Zr using chelate (DFO), and not only was the long-term biodistribution in tumors and major organs evaluated in the body, but the labeling stability of ^89^Zr conjugated to the membrane proteins was also evaluated through continuous tracking. *E. coli* accumulated at high levels in the tumor within 5 min (initial time) after tail intravenous injection, and when observed after 6 days, ^89^Zr-DFO on the surface of *E. coli* was found to be stable for a long period of time in the body. In this study, we demonstrated the long-term biodistribution and tumor-targeting effect of an *E. coli*-based drug delivery system and verified the in vivo stability of radioisotopes labeled on the surface of *E. coli*.

## 1. Introduction

Over recent decades, a range of systems for the delivery of therapeutic agents, including polymeric micro/nanoparticles, liposomes, and magnetic-based nanoparticles, have been developed and assessed [1,2,3]. These systems, which rely on either active or passive targeting mechanisms, have undergone rigorous testing in animal models and clinical trials. However, a persistent issue is their nonspecific distribution throughout the bloodstream, which can lead to significant toxicity and undesirable side effects [4]. Hence, it is crucial to develop advanced drug delivery systems that can bypass physical and biological barriers, ensuring that cancerous tissues receive adequately high drug concentrations. Bacteria utilize their cellular chemical energy to drive their helical, flagellum-fueled bio-motors, enabling fluidic propulsion. These living, motile bacteria, equipped with surface appendages like flagella and pili, display a variety of movement behaviors. These include swimming, swarming, twitching, gliding, and sliding, which permit them to navigate both liquid and semi-solid environments [5,6,7]. The process of bacterial cells actively moving toward more favorable conditions is known as bacterial taxis. This autonomous response to environmental stimuli can be tactically leveraged to direct them to specified locations within the body, a crucial aspect in designing drug delivery systems [8,9,10]. The capacity of bacterial cells to navigate through oxygen gradients is an integral attribute that aids in identifying bacteria with tumor-targeting potential. For instance, studies have revealed that *Escherichia coli* (*E. coli*) selectively colonizes and proliferates in tumor regions due to its anaerobic properties upon introduction into the body [11,12]. Concurrently, *E. coli* present in normal tissues is progressively eliminated by the immune system over time. For bacterial-based drug delivery, chemotherapeutic, radiation therapeutic, photothermal therapeutic, and immunotherapeutic drugs are loaded to bacteria. They can bind to the bacterial surface using chemical linkers such as cis-aconitic anhydride or bacterial-binding antibody interactions, and the drug can be contained inside the bacteria using bacterial metabolism. In addition, drug-loaded nanoparticles can be chemically bound to the bacterial surface or can conjugate through surface charge and complementary noncovalent interactions to enhance drug loading efficiency [6]. *E. coli* DH5α is a genetically modified non-invasive strain with the unique capability to absorb exogenous gene products and an immunity to hydrolysis by endonucleases. These attributes render *E. coli* DH5α as an excellent candidate for genetic modifications and targeted delivery [13,14]. The methodology we propose hinges on the ability of primary amine groups present on the surface of Gram-negative bacterial cells, such as *E. coli*, to form covalent bonds with various compounds that contain carboxyl or isothiocyanate groups [15]. This feature lays the foundation for the development of a robust targeted delivery system. Non-invasive imaging techniques such as positron emission tomography (PET) have made significant strides and play a pivotal role in the field of personalized medicine [16,17]. These advancements offer practical solutions to a range of currently incurable diseases, including cancer. Recently, there has been growing interest in the use of emerging radioactive isotopes including zirconium-89 (^89^Zr, half-life = 78.4 h) [18,19,20], copper-64 (^64^Cu, half-life = 12.7 h) [21,22], yttrium-86 (^86^Y, half-life = 14.7 h) [23,24], and gallium-68 (^68^Ga, half-life = 67.6 m) [25,26] for PET. The long-term in vivo tracking of drug carriers, roughly the size of a bacterium, warrants investigation. Of these radioisotopes (Ris), ^89^Zr has been thoroughly studied and shows promise for long-term tumor diagnostic studies and in vivo distribution research [27,28]. This is attributed to the ease of its relatively long-term production, its optimal nuclear decay characteristics, and its suitable chemical binding properties. In this study, we labeled *E. coli* membrane proteins with ^89^Zr and confirmed its long-term in vivo biodistribution in tumors and major organs (Figure 1). In addition, by continuously tracking ^89^Zr, we verified the labeling stability of ^89^Zr chemically bound to *E. coli* membrane proteins and we report the excellent in vivo stability of ^89^Zr-labeled *E. coli*.

## 2. Results and Discussion

### 2.1. Synthesis

The binding conditions between the primary amine groups of *E. coli* membrane proteins and p-NCS-Bz-DFO were analyzed through quantitative analysis using the fluorescent molecule FITC, which has an isothiocyanate (-NCS) group at its terminal. As a result of analyzing the reaction time, as shown in Figure 1, the highest binding rate was observed when the reaction was performed for 8 h. Nevertheless, we adopted a 1 h reaction time as this was advantageous for preserving the radioactivity of ^89^Zr and minimizing stress on the chemical reaction of *E. coli*. The FITC and *E. coli* were confirmed to bind to 65% of the maximum binding amount after 1 h.

The morphological damage and viability of *E. coli* were evaluated after sequential reactions of p-NCS-Bz-DFO and Zr. As shown in Figure 2, *E. coli* that did not participate in the chemical reaction (Figure 2a) and Zr-DFO-conjugated *E. coli* (Figure 2b) were morphologically identical. Since Zr-DFO-conjugated *E. coli* did not shrink or tear morphologically, the chemical reactions of the two steps did not affect the morphology of *E. coli*. Additionally, it was confirmed, as seen in Figure 3a, that more than 75% of Zr-DFO-conjugated *E. coli* survived.

In Figure 4a, as a result of evaluating the radiochemical purity of ^89^Zr-labeled *E. coli* purified by means of centrifugation after ^89^Zr labeling using a radio-thin-layer chromatography (TLC) scanner, a high radiochemical purity of 99.9% was confirmed. This high radiochemical purity was also confirmed when the instant thin-layer chromatography (i-TLC) plate was imaged using a PET scan (Figure 4b).

In the in vitro stability evaluation conducted using PBS and HS, high stability of 95% or more was confirmed for 7 days. This high stability of ^89^Zr-labeled *E. coli* was confirmed through in vitro stability evaluation, and it is expected that long-term stability in blood can be secured, meaning that it can be used for tumor tracking and evaluating major organ distribution (Figure 3b).

### 2.2. Cellular Uptake

In the evaluation of cellular uptake for CT-26 and A549, the uptake of ^89^Zr-labeled *E. coli* into cancer cells was confirmed. CT-26 is a mouse colon cancer cell line and A549 is a human lung cancer cell line. An analysis of both cancer cells confirmed that the uptake was increased in a time-dependent manner; in the case of A549, the uptake rate was close to 25% after 24 h, and in the case of CT-26, it was confirmed that the uptake rate was more than 15%. Although there was a difference in the amount of uptake of *E. coli* depending on the cancer cell line, it was consistent that the uptake of *E. coli* continuously increased for 24 h. Although further studies are needed, this suggests that *E. coli*-based drug delivery can be effective in various cancer cells and confirms the possibility of continuous accumulation in tumors in the body (Figure 5).

### 2.3. PET Studies

PET was performed to evaluate the drug delivery effect of ^89^Zr-labeled *E. coli* in the body. ^89^Zr-labeled *E. coli* cells were infused via tail vein injection and were evaluated for 6 days (Figure 6). It was confirmed that *E. coli* cells accumulated in the cancer tissue at the initial time after tail intravenous injection, and it was found that the accumulation was maintained for 6 days. The representative reticuloendothelial system (RES), including the liver and spleen, showed high accumulation, but as shown in Figure 7, as a result of tracking the regions of interest (ROIs) of the liver, bladder, and tumor for 6 days, time-dependent clearance of ^89^Zr-labeled *E. coli* was confirmed. As no increase in excretion-related organ- and bone-specific uptake was observed over time, it is believed that there was no decomposition of *E. coli* by fasting cells of the liver and spleen. The clearance of *E. coli* to the RES in the future is worth studying intensively. The high accumulation in the liver and spleen needs to be improved to increase drug delivery efficiency. Studies that reduce toxicity to normal cells by introducing polyethylene glycol (PEG) on the bacterial surface can be referred to, and as the introduction of PEG reduces the accumulation of the RES of nanoparticles, it needs to be studied in *E. coli*-based drug delivery systems [29]. The radioactivity observed in the bladder over a period of 1 day appears to be related to the clearance of ^89^Zr-DFO dissociated from *E. coli*, considering the diameter of *E. coli* cells (1–2 µm) [30,31]. In the case of tumors, ^89^Zr-labeled *E. coli* accumulation was observed to increase, and the tumor/liver ratio was also confirmed to continuously increase for 6 days. Dissociated ^89^Zr has been reported to accumulate in bones in the body [31,32,33]. Nevertheless, bone accumulation by ^89^Zr was not observed, as seen in Figure 7, so it is not dissociated from *E. coli* after intravenous injection in the body, and this result is associated with high in vitro labeling stability (Figure 4b). In addition, *E. coli* has high potential as a drug delivery system that accumulates in tumor tissues at a very fast rate and is maintained for a long time. Eight days after intravenous injection, the mice were sacrificed to confirm the biodistribution in major organs. As shown in Figure 8, there was low accumulation in major organs except for the liver and spleen, suggesting that *E. coli*-based drug carriers can reduce side effects on normal organs.

## 3. Materials and Methods

### 3.1. Materials

Fluorescein isothiocyanate (FITC), diethylenetriaminepentaacetic acid (DTPA), dimethyl sulfoxide (DMSO), zirconium (IV) chloride, and hydrogen chloride (HCl) were purchased from Merck (Darmstadt, Germany). N1-hydroxy-N1-(5-(4-(hydroxy(5-(3-(4-isothiocyanatophenyl)thioureido)pentyl)amino)-4-oxobutanamido)pentyl)-N4-(5-(N-hydroxyacetamido)pentyl)succinamide (p-NCS-Bz-DFO) was purchased from Chematech (Dijon, France). Dulbecco’s modified Eagle’s medium (DMEM), phosphate-buffered saline (PBS), HEPES buffer solution, human serum (HS), and fetal bovine serum (FBS) were purchased from Gibco BRL Life Technologies (Waltham, MA, USA). QMA Plus Short Cartridge was purchased from Waters (Worcester County, MA, USA). Bacterial Viability Kit (LIVE/DEAD™ BacLight™ Bacterial Viability Kit, Middlesex County, MA, USA) was purchased from Thermo Fisher Scientific (Middlesex County, MA, USA). CT-26 and A549 cells were procured from the Korean cell line bank (Seoul, Republic of Korea). Balb/c mice (20 ± 1.5 g, female) were purchased from Orient Bio (Seongnam, Republic of Korea). Zirconium-89 was procured from the Korea Atomic Energy Research Institute (KAERI, Jeongeup, Republic of Korea) and produced using RFT-30 (30 MeV cyclotron). Quantitative analysis of FITC was performed using a fluorescence spectrometer (Hitachi, Tokyo, Japan). Morphological data of *E. coli* were obtained using a field-emission scanning electron microscope (FE-SEM) (Hitachi, Tokyo, Japan). Radiochemical yield was assessed using an AR-2000 radio-TLC imaging scanner (Bioscan, Santa Barbara, CA, USA). The in vitro stability of the radiolabeled sample was measured using a Wizard-1470 automatic gamma counter (Perkin Elmer, Waltham, MA, USA). Small animal PET imaging was carried out using a Genesis 4 (Sofie Biosciences, Culver City, CA, USA).

### 3.2. Preparation of FITC-Conjugated E. coli

One milligram of FITC was placed in a 10 mL vial and dispersed in 0.1 mL of DMSO. Then, 9.9 mL of 0.1 M HEPES (pH 8) was added into the vial and stirred for 10 min. After stirring, *E. coli* (1.0 × 10^9^ cells) was added and incubated for 1 h at 25 °C. The solvent and FITC-labeled *E. coli* were separated by means of centrifugation and washed two times with physiological saline to remove unreacted FITC. Centrifugation was carried out at 500 rcf for 5 min. FITC-conjugated *E. coli* was measured with a fluorescence spectrometer and the calibration curve was used to quantify FITC bound to the *E. coli* surface.

### 3.3. Preparation of Zr-Conjugated E. coli

One milliliter of *E. coli* (2 × 10^8^/mL in 0.1 M HEPES) was reacted with 80 nmol of p-NCS-Bz-DFO for 1 h, followed by washing twice with PBS. Then, 1 mL of 0.1 M HEPES (pH 8) was reacted with 80 nmol of ZrCl_4_ for 1 h. After the reaction, *E. coli* bacteria were washed twice with 50 mM DTPA to remove free Zr. Centrifugation was carried out at 500 rcf for 5 min each time. For the viability evaluation of *E. coli* after the chemical reaction, the activity of *E. coli* was measured using a bacterial viability kit.

### 3.4. Preparation of ^89^Zr-Labeled E. coli

^89^Zr(ox)_2_ was exchanged for ZrCl_4_ using a QMA cartridge and 1M HCl aqueous solution was used as the elution solvent. The reaction was performed at room temperature for 1 h by adding ^89^ZrCl_4_ of 37 MBq/10 μL to a 2 mL tube containing 1 μg of *p*-NCS-Bz-NCS. Then, a 1 mL *E. coli* dispersion (1 × 10^9^/mL in 0.1 M HEPES, pH8) was added to the reaction vial. After the completion of the reaction, the solvent and ^89^Zr-labeled *E. coli* were separated using centrifugation, washed with a 50 mM DTPA (pH 7) solution to remove free ^89^Zr, and washed two more times with PBS. Centrifugation was performed at 500 rcf for 5 min each time.

### 3.5. In Vitro Stability

^89^Zr-labeled *E. coli* bacteria were dispersed in PBS to test their in vitro stability. Then, 100 μL of *E. coli*, labeled with 37 MBq/mL, was added to 1 mL of 10% HS and 10% FBS. The mixture in the tube was carefully stirred at 37 °C for 7 days. The in vitro stability was assessed at specific time intervals (1, 2, and 4 h and 1, 2, 5, and 7 days) using a radio-TLC scanner. For the chromatographic process, the mobile phase utilized 50 mM of DTPA, while the stationary phase employed i-TLC-SG with dimensions of 1.0 × 10 cm.

### 3.6. Cellular Uptake

For cellular uptake, 1.0 × 10^5^ cancer cells (CT-26 and A549) were seeded into 24-well plates and grown in DMEM medium containing 10% (*v*/*v*) FBS at 37 °C for 24 h. ^89^Zr-labeled *E. coli* was added with 190 kBq/well to cancer cells and incubated for 48 h at 37 °C. The sample was washed two times with PBS and cell-associated radioactivity was detected using a gamma ray counter at 0.5, 1, 4, and 24 h.

### 3.7. PET Studies and Ex Vivo Biodistribution

PET studies were conducted in accordance with animal testing guidelines and ethics and were approved by the Korea Atomic Energy Research Institute (IACUC KAERI-2023-002). PET images were taken to determine the in vivo distribution of ^89^Zr-labeled *E. coli*. CT-26 cells were dispersed in saline and injected subcutaneously into the thighs of Balb/c mice (5.0 × 10^6^ per mouse). After the length of tumors had grown to 120 mm, ^89^Zr-labeled *E. coli* was injected intravenously into the tail of each mouse (3.7 MBq/100 µL per mouse). As for the injection dose, 3.7 MBq was selected to effectively track Zr-89 for more than 6 days. The mice were anesthetized with 2% isoflurane and whole-animal imaging was measured using PET for 6 days. Mice were measured by designating a time point of 30 min, 1, 2, 4, 6, and 24 h, and 2, 5, and 6 days, and the time required for the measurement was approximately 5 min. Then, the mice were sacrificed and ex vivo biodistribution in major organs was measured using PET and a gamma counter. PET studies were analyzed using Amide’s Medical Image Data Examiner (AMIDE, open-source software tool, version 1.0.6). Using PET image reconstruction and quantification conditions, the maximum threshold was 10 and the minimum threshold was 0. The voxel size was 0.45 × 0.45 × 0.45 mm. The quantitative values were calculated using a maximum SUV.

## 4. Conclusions

In this study, we confirmed the remarkable effectiveness and in vivo stability of an *E. coli*-based drug carrier for cancer diagnosis and treatment using the radioisotope ^89^Zr. *E. coli* accumulated in the tumor quickly (within 5 min) after intravenous injection and remained accumulated for 6 days. Additionally, ^89^Zr-labeled *E. coli* cells were confirmed to be stable for 7 days in human serum. Since the accumulation of free ^89^Zr in bone was not observed when it was administered, it was confirmed that ^89^Zr was specifically bound to the chelate (DFO). Considering that clearance through the bladder did not increase after 24 h, ^89^Zr-DFO on the surface of *E. coli* was stable for a long period of time in the body. In conclusion, this study can contribute to research on the in vivo tracking of various Gram-negative bacteria with tumor therapeutic effects and to the development of a tumor-targeting drug delivery system that can overcome physical and biological barriers.

## Data Availability

The data presented in this study are available upon request from the corresponding author.

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
