# Peer review of "Long-Term Tumor-Targeting Effect of E. coli as a Drug Delivery System"

_pharmaceuticals, 2024, doi:10.3390/ph17040421_

Round 1

Reviewer 1 Report

Comments and Suggestions for Authors

The article is dedicated to the development and the pre-clinical evaluation of the 89Zr-labeled E. coli. The authors show and describe in detail all procedures for precursor, cold reference and radiotracer synthesis. There are a small number of typographical errors in the paper. The following comments are made about the paper:

1) In section 2.1, the authors should add a discussion of the radiolabeling procedure. Although the direct method is more efficient (according to the literature), I understand from the paper that the authors chose the indirect method of radiolabeling. Therefore, it is not clear why this is the preferred method for this substrate. In addition, it would be interesting to know the value of the radiochemical yield and the radiochemical conversion of this method.

2) In sections 3.2-3.4, the authors should specify the pH range. The authors most likely used HEPES buffer. The pH of HEPES as an acid is about 5.5, and the conjugation of NCS and NH2 occurs at pH > 8.

3) The most common form of 89Zr is 89Zr oxalate, so a method for the preparation of 89Zr chloride is not described.

4) Has the stability (especially morphology) of FITC/Zr-labeled E. coli in PBS and HS been investigated?

However, these comments do not diminish the quality of the paper. The article may be accepted

for publication after minor revisions.

Author Response

Dear reviewer,

Thank you for the great revision.

Our manuscript has been refined due to your comments. It has also helped us a lot in our ongoing research. 

I hope you and your institute will continue to prosper.

Thank you.

Warm regards,

Gun Gyun Kim

Response to Reviewer 1 Comments

1)  In section 2.1, the authors should add a discussion of the radiolabeling procedure. Although the direct method is more efficient (according to the literature), I understand from the paper that the authors chose the indirect method of radiolabeling. Therefore, it is not clear why this is the preferred method for this substrate. In addition, it would be interesting to know the value of the radiochemical yield and the radiochemical conversion of this method.

Response 1: Thank you for your great opinion. Nevertheless, I wonder if we understand the reviewer's opinion properly. We used the fluorescent material FITC to quantify how many isothiocyanate (-NCS) groups bind to the primary amine groups of E. coli membrane proteins. For 89Zr labeling, an isothiocyanate group was subsequently labeled 89Zr to DFO (p-NCS-Bz-NCS) present at the terminal and bound to the surface of E. coli. DFO is the most widely used for 89Zr labeling, and its high labeling stability has also been demonstrated.

2)  In sections 3.2-3.4, the authors should specify the pH range. The authors most likely used HEPES buffer. The pH of HEPES as an acid is about 5.5, and the conjugation of NCS and NH2 occurs at pH > 8.

Response 2: Thank you for the great comment from our reviewer. The pH of the HEPES buffer we used is pH 8. I specified the value of pH in the manuscript. (Line 203, 212, 220)

3)  The most common form of 89Zr is 89Zr oxalate, so a method for the preparation of 89Zr chloride is not described.

Response3: The experimental method was added according to the reviewer's opinion. (Line 203, 212, 220)

4)  Has the stability (especially morphology) of FITC/Zr-labeled E. coli in PBS and HS been investigated?

Response4: Thank you for your excellent opinion. Morphology after Zr-labeled E. coli reaction was confirmed using SEM, but morphology stability after exposure to PBS and HS was not confirmed. Instead, in Fig. 4b, the labeling stability of 89Zr-labeled E. coli in PBS and HS was confirmed that 89Zr was stable long enough not to dissociate.

Reviewer 2 Report

Comments and Suggestions for Authors

Accepted. 

Comments on the Quality of English Language

It needs minor modification. 

Author Response

Dear reviewer,

Thank you for the great revision.

Our manuscript has been refined due to your comments.

It has also helped us a lot in our ongoing research.

Thank you.

Warm regards,

Gun Gyun Kim

Reviewer 3 Report

Comments and Suggestions for Authors

The manuscript entitled “Long-term tumor-targeting effect of 89Zr-labeled e.coli as drug delivery system” describes the radiolabeling of e.coli with zirconium-89 and the assessment of the radiolabeled bacteria’s pharmacokinetics and tumor uptake in murine tumor models by PET imaging to determine the feasibility of e.coli as drug delivery system.

The manuscript requires some additional details (as major corrections) and some minor adjustments to be considered for publication:

Major:

A)   The use of nanoparticles and liposomes as drug delivery systems has been widely described and multiple chemistries and applications can be found in the literature. E. coli as drug-delivery vector is something quite novel and, understandingly, not much literature can be found on the subject to get an idea of its potentiality. Can the authors add a small paragraph (in Introduction) indicating what type of drug could be delivered using e.coli and how the drugs would be loaded onto the bacteria (i.e., by same chemical modifications and functionalities as the ones used on other nano/micro particles? Would the authors investigate a more targeted tumor delivery of the e.coli bacteria?). Also, what would be the benefits and the limitations on this kind of delivery system?

B)  Page 3, line 108: The authors mention “internalization” of the radiolabeled e.coli but did not show any experiment to confirm it. Either add the results of a suitable internalization assay (which is something that would add value to the manuscript) or remove or amend the whole sentence to avoid mentioning something that has not been experimentally confirmed.

 C) Page 3, line 115: A tumor uptake time-activity curve would be informative. Also, since quite high, the uptake in the liver is rightly included in Fig 7, but the SUV values of the spleen, which is the other main organ with high uptake, are missing and should be included for completeness, if possible. Additionally, a table with the biodistribution ID/g% values (to complement Fig. 8, maybe in Supplemental Information?) would be useful too. Can the authors add a line about probable reasons why the radiolabeled e.coli accumulate in (and eventually clear from) the liver and the spleen?

 D) Page 3, line 127: The authors state that the lack of bone radioactivity uptake is a consequence of the high stability of the 89Zr-e.coli construct. The statement might be right but it is just a speculation and needs more evidence. Surely the authors agree that it is generally accepted that bone uptake is linked to the demetallation of 89Zr-DFO-based agents. Such demetallation is the consequence of the non-complete coordination of zirconium-89 by the hexadentated chelator (DFO) and not of the vector it is attached to. Based on that, the authors should have some degrees of demetallation and therefore bone uptake due to the simple fact that they use DFO to coordinate zirconium-89, independently from the vector. This is something that the authors did not mention and should be mentioned in the manuscript. Said that, the fact that the images and the biodistribution data do not show the expected high bone uptake is interesting and the influence of the e.coli itself is a possible option. However, it is something that requires further investigation and should not be given as an accepted reason. If the authors have additional experimental proof supporting their current statement, they should include it in the manuscript. If the authors have not performed any extra experiments, this part of Discussion should be amended.

 E) Page 3, PET studies: The 89Zr-e.coli construct gets to the tumor quickly (after just 5 min p.i.) but there is also high uptake in non-target organs like liver and spleen. Would that be an issue for a drug delivery system? How would the authors reduce the liver and spleen uptake and improve the tumor uptake to improve the drug delivery efficiency (and avoid effects in healthy tissues)? Some discussion on this subject should be added to the manuscript.

 F) Page 7, PET studies: The PET quantification method is missing and should be added in Material and Methods. A table with the SUV values would be very informative too (maybe in Supplemental Information?).

Minor:

The English throughout the whole manuscript (and abstract) needs improvement. Some whole sentences are not clear and difficult to understand.

Title: “Long-term tumor-targeting effect of 89Zr-labeled e.coli as drug delivery system” gives the impression that the authors want to use zirconium-89-labeled e.coli as delivery system and not just the bacterial itself (without the radioactive label). Is that the case? If not, please amend the title accordingly.

Page 2, line 67: The abbreviation RIs has not been explained.

Scheme 1 (page 2) is not cited in the main text and looks like a graphical abstract. What did the authors wanted to use scheme 1 for?

Page 2, 2.1. Synthesis: How many DFO (of FITC) molecule per bacteria did the authors consider suitable for their work? Why the 4000 DFO/bacterium achieved after 5 min incubation were not enough? The author should add few lines about the reason and benefits that lead the authors to choose to produce bacteria with such high number of chelators (i.e., 8000 DFO/FITC).

Page 3, line 88: The authors should add a line describing the morphological changes that any damage would have created on the bacteria. What would the authors consider as morphological changes/damages?

Page 3, lines 98-101: This section discusses figure 4b but the citation of the figure is missing. Please add.

Page 3, 2.2. Cellular uptake: Why did the authors chose the CT-26 and A549 cell lines amongst the very wide accessible selection of cells. Based on their availability or characteristics? The authors should add a short line explaining the reason for the choice of cell lines.

Page 3, line 111: With the PET studies, the authors did not evaluate the drug delivery effect of 89Zr-labeled e.coli in the body. The authors assessed the pharmacodynamics and tumor uptake of 89Zr-labeled e.coli. Please change the sentence accordingly.

Page 3, PET studies: Can the authors add one short line explaining the reasons for the choice of the 89Zr-e.coli injected dose (3.7 MBq and unidentified quantity of bacteria)?

Figure 8. Error bars are missing. How many mice were biodistributed?

Page 6, line 169: The authors mention MDA-MB-23 cell line instead of A549.

Page 7, line 180: How was the number of FITC attached to e.coli estimated? Please add the missing details of the technique used.

Page 7, line 186, 192 and 200: The “rpm” of a centrifuge depends on its rotor. Can the authors indicate the “rcf” (or “g”) value which is independent from the rotor dimensions? Alternatively, the authors can give the details of the centrifuge they used.

Page 7, line 212: “Cell associated radioactivity” should be used instead of “89Zr labelled e.coli uptake rate” since the experiment performed is not measuring rates.

Page 7, line 220: The authors inject 3.7MBq of agent in the mice. Just for completion, is it possible to estimate how many e.coli cells were injected?

Page 8, line 222: The authors state that PET images were performed up to 7 days p.i. but the PET images shown in figure 6 are up to 6 days. Is there a missing image?

Comments on the Quality of English Language

The English throughout the whole manuscript (and abstract) needs improving. Some whole sentences are not clear and difficult to understand.

Author Response

Dear reviewer,

Thank you for the great revision.

Our manuscript has been refined due to your comments. It has also helped us a lot in our ongoing research. 

I hope you and your institute will continue to prosper.

Thank you.

Warm regards,

Gun Gyun Kim

Response to Reviewer 3 Comments

Major

A) The use of nanoparticles and liposomes as drug delivery systems has been widely described and multiple chemistries and applications can be found in the literature. E. coli as drug-delivery vector is something quite novel and, understandingly, not much literature can be found on the subject to get an idea of its potentiality. Can the authors add a small paragraph (in Introduction) indicating what type of drug could be delivered using e.coli and how the drugs would be loaded onto the bacteria (i.e., by same chemical modifications and functionalities as the ones used on other nano/micro particles? Would the authors investigate a more targeted tumor delivery of the e.coli bacteria?). Also, what would be the benefits and the limitations on this kind of delivery system?

Response A : Thank you for the reviewer's great comments. Content was added in response to the reviewer's opinion (Line 50-56). As the reviewer's opinion, there are significantly fewer studies on bacterial-based drug delivery systems than on nanoparticle-based studies. However, considering the tumor target effect of E. coli, it is a field that is expected to develop through continuous improvement research. The high accumulation in RES and the lack of research on potential toxicity suggest that steady research on E. coli-based drug delivery systems is needed in the future.

B) Page 3, line 108: The authors mention “internalization” of the radiolabeled e.coli but did not show any experiment to confirm it. Either add the results of a suitable internalization assay (which is something that would add value to the manuscript) or remove or amend the whole sentence to avoid mentioning something that has not been experimentally confirmed.

Response B: Thank you for your great opinion. I modified the contents according to the reviewer's opinion. (Line 113-117)

C) Page 3, line 115: A tumor uptake time-activity curve would be informative. Also, since quite high, the uptake in the liver is rightly included in Fig 7, but the SUV values of the spleen, which is the other main organ with high uptake, are missing and should be included for completeness, if possible. Additionally, a table with the biodistribution ID/g% values (to complement Fig. 8, maybe in Supplemental Information?) would be useful too. Can the authors add a line about probable reasons why the radiolabeled e.coli accumulate in (and eventually clear from) the liver and the spleen?

Response C: Thank you for your great opinion, I added the content (Line 129-132) and revised the Fig. 7 according to the reviewer's opinion.

D) Page 3, line 127: The authors state that the lack of bone radioactivity uptake is a consequence of the high stability of the 89Zr-e.coli construct. The statement might be right but it is just a speculation and needs more evidence. Surely the authors agree that it is generally accepted that bone uptake is linked to the demetallation of 89Zr-DFO-based agents. Such demetallation is the consequence of the non-complete coordination of zirconium-89 by the hexadentated chelator (DFO) and not of the vector it is attached to. Based on that, the authors should have some degrees of demetallation and therefore bone uptake due to the simple fact that they use DFO to coordinate zirconium-89, independently from the vector. This is something that the authors did not mention and should be mentioned in the manuscript. Said that, the fact that the images and the biodistribution data do not show the expected high bone uptake is interesting and the influence of the e.coli itself is a possible option. However, it is something that requires further investigation and should not be given as an accepted reason. If the authors have additional experimental proof supporting their current statement, they should include it in the manuscript. If the authors have not performed any extra experiments, this part of Discussion should be amended.

Response D: Thank you for your great opinion. I've revised the content according to the reviewer's opinion. (Line 144-148)

E) Page 3, PET studies: The 89Zr-e.coli construct gets to the tumor quickly (after just 5 min p.i.) but there is also high uptake in non-target organs like liver and spleen. Would that be an issue for a drug delivery system? How would the authors reduce the liver and spleen uptake and improve the tumor uptake to improve the drug delivery efficiency (and avoid effects in healthy tissues)? Some discussion on this subject should be added to the manuscript.

Response E: Thank you for your great opinion. I've added content based on the reviewer's comments. (Line 132-136)

F) Page 7, PET studies: The PET quantification method is missing and should be added in Material and Methods. A table with the SUV values would be very informative too (maybe in Supplemental Information?).

Response F: Thank you for your great opinion. We added the software we used for PET quantification based on the reviewer's opinion. (Line 288)

Minor

1) The English throughout the whole manuscript (and abstract) needs improvement. Some whole sentences are not clear and difficult to understand.

Response 1: I edited my English using the English Editing service provided by MDPI (English Editing ID: English-77557)

2) Title: “Long-term tumor-targeting effect of 89Zr-labeled e.coli as drug delivery system” gives the impression that the authors want to use zirconium-89-labeled e.coli as delivery system and not just the bacterial itself (without the radioactive label). Is that the case? If not, please amend the title accordingly.

Response 2: Thank you for your great opinion, I've revised it according to the reviewer's opinion.

3) Page 2, line 67: The abbreviation RIs has not been explained.

Response 3: Thank you for your great opinion. I modified the title based on the reviewer's opinion. (Line 71)

4) Scheme 1 (page 2) is not cited in the main text and looks like a graphical abstract. What did the authors wanted to use scheme 1 for?

Response 4: Thank you for the reviewer's great comments. Scheme 1 has been inserted to support the introduction and we've added citations to the manuscript. (Line 76)

5) Page 2, 2.1. Synthesis: How many DFO (of FITC) molecule per bacteria did the authors consider suitable for their work? Why the 4000 DFO/bacterium achieved after 5 min incubation were not enough? The author should add few lines about the reason and benefits that lead the authors to choose to produce bacteria with such high number of chelators (i.e., 8000 DFO/FITC).

Response 4: Thank you for your great opinion. In order to minimize the potential side effects caused by E. coli, I tried to minimize the number of bacteria injected into the body. To do this, the reaction time was analyzed for binding of as much as possible amount of DFO (of FITC) per bacterium.

6) Page 3, line 88: The authors should add a line describing the morphological changes that any damage would have created on the bacteria. What would the authors consider as morphological changes/damages?

Response 6: Thank you for your great comment. If E. coli is damaged by exposure to chemical reactions, morphological shriveling and tearing deformation may occur. I wanted to measure the SEM to confirm that the E. coli morphology did not change.

7) Page 3, lines 98-101: This section discusses figure 4b but the citation of the figure is missing. Please add.

Response 7: Thank you for the reviewer's great comments. I've added content based on the reviewer's comments. (Line 106)

8) Page 3, 2.2. Cellular uptake: Why did the authors chose the CT-26 and A549 cell lines amongst the very wide accessible selection of cells. Based on their availability or characteristics? The authors should add a short line explaining the reason for the choice of cell lines.

Response 8: Thanks for the reviewer's excellent opinion. Since there are no reports of E. coli having a targeted effect on certain cancer cells, I wanted to arbitrarily select and compare mouse cancer cell lines and human cancer cell lines.

9) Page 3, line 111: With the PET studies, the authors did not evaluate the drug delivery effect of 89Zr-labeled e.coli in the body. The authors assessed the pharmacodynamics and tumor uptake of 89Zr-labeled e.coli. Please change the sentence accordingly.

Response 9: Thank you for the reviewer's great comments. I've revised the content that may confuse readers according to the reviewer's comments. (Line 78, 106, 258)

10) Page 3, PET studies: Can the authors add one short line explaining the reasons for the choice of the 89Zr-e.coli injected dose (3.7 MBq and unidentified quantity of bacteria)?

Response 10: Thank you for your good opinion, and I've added content based on the reviewer's comments. (Line 248)

11) Figure 8. Error bars are missing. How many mice were biodistributed?

Response 11: Thank you for your good opinion. For the Ex Vivo evaluation, I sacrificed 1 mouse. Corrected incorrectly written content. (Line 180, 252)

12) Page 6, line 169: The authors mention MDA-MB-23 cell line instead of A549.

Response 12: Thank you for your good opinion, I modified the content according to the reviewer's opinion. (Line 193)

13) Page 7, line 180: How was the number of FITC attached to e.coli estimated? Please add the missing details of the technique used.

Response 13: Thank you for your great advice. We used a fluorescence spectrometer to quantify the fluorescence intensity of FITC. I added it to the experimental method. (Line 210)

14) Page 7, line 186, 192 and 200: The “rpm” of a centrifuge depends on its rotor. Can the authors indicate the “rcf” (or “g”) value which is independent from the rotor dimensions? Alternatively, the authors can give the details of the centrifuge they used.

Response 14: Thank you for your great opinion. I modified it according to the reviewer's opinion. (Line 210, 218, 228)

15) Page 7, line 212: “Cell associated radioactivity” should be used instead of “89Zr labelled e.coli uptake rate” since the experiment performed is not measuring rates.

Response 15: Thank you for your great opinion. I modified it according to the reviewer's opinion. (Line 240)

16) Page 7, line 220: The authors inject 3.7MBq of agent in the mice. Just for completion, is it possible to estimate how many e.coli cells were injected?

Response 16: Thank you for your great opinion. However, when I selected the injection amount, I used the amount of radiation as the standard. I haven't confirmed the exact amount of bacteria, but I will apply it to future studies.

17) Page 8, line 222: The authors state that PET images were performed up to 7 days p.i. but the PET images shown in figure 6 are up to 6 days. Is there a missing image?

Response 17: Thank you for your great comments. There was a mistake in the process of writing the manuscript. I revised it to match Figure 6. (Line 251)

Reviewer 4 Report

Comments and Suggestions for Authors

Dr. Gun Gyun Kim and colleagues report to show the capability of 89Zr-labeled E. Coli as tumor imaging PET agent. This provides some interesting insights, however there are several comments on this manuscript as follows.

1)    In the Title, “E. Coli” should be corrected to “E. coli”, because it is the scientific name.

2)    Throughout the manuscript, all description of “E. coli” should be an italic form. In addition, “Escherichia coli” in Keywords should also be an italic form.

3)    When the term of “E. coli” is firstly appeared in either Abstract or Introduction, it should be shown as “Escherichia coli (E. coli)”.

4)    In Abstract and several parts in the manuscript, authors repeatedly describe regarding the capability of E. coli as a carrier of drug delivery system. As shown in this manuscript, radio-labeled E. coli may be useful for a tool of tumor-imaging agent. However, apart from nanoparticles and liposomes, it is doubtful how E. coli can load to carry the therapeutic drugs to tumor tissue, even if it can target to the tumor tissue.

5)    In Abstract and several parts in the manuscript, authors repeatedly mention that “89Zr-labeled E. Coli was rapidly uptake in the tumor within 5 min after the intravenous injection”. Do the “Initial” in Figure 6 and “Ini” in Figure 7 mean the uptake from 0 to 5 min post injection? If so, authors should explain more clearly.

6)    Related to my comments 4 and 5, if E. coli is applicable for tumor imaging agent showing very fast uptake into the tumor, but not for the carrier of drug delivery system, the positron emitters with shorter half-lives, like C-11 and F-18, should be more suitable. Positron emitters with longer half-lives like 89Zr cause induce more radiation exposure dose to the patients.

7)    In 2. Results and Discussion, 2.1. Synthesis, at page 3, line 95, “TLC” should be spelled out. In addition, it should be explained what “i-TLC” is.

8)    In 2. Results and Discussion, 2.2. Cellular Uptake, at page 3, line 103, the terms of “CT-26” and “A549” suddenly appear. It may be better to describe here what kinds of tumor cells they are. The potential readers cannot recognize them until the legend of Figure 5.

9)    In 2. Results and Discussion, 2.3. PET Studies, at page 3, line 115, “RES” should be spelled out.

10) In 2. Results and Discussion, 2.3. PET Studies, at page 3, line 116, “region of interest (ROI)” should be corrected to “regions of interest (ROIs)”. In contrast, at page 3, line 120, regions of interest (ROIs)” should be corrected to “ROIs”.

11) In 2. Results and Discussion, 2.3. PET Studies, at page 3, lines 115-118 and lines 119-122 repeatedly indicate almost similar content. It should be changed to more concise description.

12) In the right Y axis of Figure 7, authors indicate the “Tumor/Liver ratio”, showing over 40 at 6-day post 89Zr-labeled E. Coli injection. However, it seems strange that the uptakes in the liver are always higher than those in the tumor from the initial to 6-days post. This is also demonstrated by ex vivo analysis shown in Figure 8. Author should explain how to interpret from the absolute uptake values in the liver and tumor to their ratios.

13) In the legend of Figure 8, it seems strange that the legend describes something rerated PET measurement, but data demonstrate ex vivo analysis.

14) In 3. Materials and Methods, 3.7. PET Studies and Ex Vivo Biodistribution, at page 7, line 219, what does it mean “120mm” for tumor tissue size? It is usually considered the tumor volume (mm3) when apply for PET imaging.

15) In 3. Materials and Methods, 3.7. PET Studies and Ex Vivo Biodistribution, at page 7, line 221, authors describe the protocol for PET measurements. More detailed information should be added; how long measured in the first PET scanning (up to 6 hours?) followed by how many times repeated in same animals.

Author Response

Dear reviewer,

Thank you for the great revision.

Our manuscript has been refined due to your comments. It has also helped us a lot in our ongoing research. 

I hope you and your institute will continue to prosper.

Thank you.

Warm regards,

Gun Gyun Kim

Response to Reviewer 4 Comments

1. In the Title, “E. Coli” should be corrected to “E. coli”, because it is the scientific name.

Response 1: I modified the title based on the reviewer's opinion.

2. Throughout the manuscript, all description of “E. coli” should be an italic form. In addition, “Escherichia coli” in Keywords should also be an italic form.

Response 2:  I revised it according to the reviewer's opinion.

3. When the term of “ coli” is firstly appeared in either Abstract or Introduction, it should be shown as “Escherichia coli(E. coli)”.

Response 3: I revised it according to the reviewer's opinion.

4. In Abstract and several parts in the manuscript, authors repeatedly describe regarding the capability of  colias a carrier of drug delivery system. As shown in this manuscript, radio-labeled E. coli may be useful for a tool of tumor-imaging agent. However, apart from nanoparticles and liposomes, it is doubtful how E. coli can load to carry the therapeutic drugs to tumor tissue, even if it can target to the tumor tissue.

Response 4: Thank you for your comment. Labeling of Zr-89 is not essential when using E. coli as a drug carrier. So, you can either directly bind drug molecules to surface membrane proteins or chemically bind drug-supported liposomes. More details have been added to this text. (Line 50-56)

5. In Abstract and several parts in the manuscript, authors repeatedly mention that “89Zr-labeled  Coliwas rapidly uptake in the tumor within 5 min after the intravenous injection”. Do the “Initial” in Figure 6 and “Ini” in Figure 7 mean the uptake from 0 to 5 min post injection? If so, authors should explain more clearly.

Response 5: Information about the initial time is clearly indicated in the abstract part. (Line 21)

6. Related to my comments 4 and 5, if  coliis applicable for tumor imaging agent showing very fast uptake into the tumor, but not for the carrier of drug delivery system, the positron emitters with shorter half-lives, like C-11 and F-18, should be more suitable. Positron emitters with longer half-lives like 89Zr cause induce more radiation exposure dose to the patients.

Response 6: I completely agree with the reviewer. The reason for labelling Zr-89 was to check the behaviours and tumor accumulation effect of E. coli in the body for a long time and to evaluate the stability of the label in the body of the membrane protein-mediated radioisotope. Short-half-life nuclides would be preferred for the diagnosis of tumors, and the reviewer suggested C-11 and F-18 are excellent candidates, and Ga-68 (metals, half-life: 68 min) are examples of nuclides that can apply similar labelling methods to this study. Thank you for the great comment.

7. In 2. Results and Discussion, 1. Synthesis, at page 3, line 95, “TLC” should be spelled out. In addition, it should be explained what “i-TLC” is.

Response 7: I revised it according to the reviewer's opinion. (Line 99, 101)

8. In 2. Results and Discussion, 2. Cellular Uptake, at page 3, line 103, the terms of “CT-26” and “A549” suddenly appear. It may be better to describe here what kinds of tumor cells they are. The potential readers cannot recognize them until the legend of Figure 5.

Response 8: I revised it according to the reviewer's opinion. (Line 109)

9. In 2. Results and Discussion, 3. PET Studies, at page 3, line 115, “RES” should be spelled out.

Response 9: I revised it according to the reviewer's opinion. (Line 126)

10. In 2. Results and Discussion, 2.3. PET Studies, at page 3, line 116, “region of interest (ROI)” should be corrected to “regions of interest (ROIs)”. In contrast, at page 3, line 120, “regions of interest (ROIs)” should be corrected to “ROIs”.

Response 10: I revised it according to the reviewer's opinion. (Line 128)

11. In 2. Results and Discussion, 2.3. PET Studies, at page 3, lines 115-118 and lines 119-122 repeatedly indicate almost similar content. It should be changed to more concise description.

Response 11: I revised it according to the reviewer's opinion. (Line 129-133)

12. In the right Y axis of Figure 7, authors indicate the “Tumor/Liver ratio”, showing over 40 at 6-day post 89Zr-labeled  Coliinjection. However, it seems strange that the uptakes in the liver are always higher than those in the tumor from the initial to 6-days post. This is also demonstrated by ex vivo analysis shown in Figure 8. Author should explain how to interpret from the absolute uptake values in the liver and tumor to their ratios.

Response 12: Thank you for the great comment, I have calculated the value in the way below.

[Tumor SUVmax / Liver SUVmax x 100]

13. In the legend of Figure 8, it seems strange that the legend describes something rerated PET measurement, but data demonstrate ex vivo analysis.

Response 13: I revised it according to the reviewer's opinion. (Line 176)

14. In 3. Materials and Methods, 3.7. PET Studies and Ex Vivo Biodistribution, at page 7, line 219, what does it mean “120mm” for tumor tissue size? It is usually considered the tumor volume (mm3) when apply for PET imaging.

Response 14: It represents the maximum diameter of the tumor, and the volume of the tumor could not be measured. In future studies, we will measure the volume of the tumor according to the reviewer's opinion.

15.  In 3. Materials and Methods, 3.7. PET Studies and Ex Vivo Biodistribution, at page 7, line 221, authors describe the protocol for PET measurements. More detailed information should be added; how long measured in the first PET scanning (up to 6 hours?) followed by how many times repeated in same animals.

Response 15: We added PET measurement protocol according to reviewer's opinion. (Line 246-248)

Round 2

Reviewer 3 Report

Comments and Suggestions for Authors

The authors response to the comments is mostly satisfactory. Just the following minor observations:

Correction in connection to Comment F) (PET quantification): Firstly, the software used for the PET analysis and quantification is important and should be included in the highly visible PET section in Material and Methods and not in the "Editing Software" section (which is hidden at the end of the manuscript and in small print). Secondly, a minimal description of the performed scanning procedure and the quantification is missing and should be added in the PET section in Material and Method (including any type of corrections performed or not, how were the images reconstructed? voxel size? What type of SUV was calculated (SUVmean, max, peak) and how?).

The authors' answer to Comment 6) (i.e., E. coli is damaged by exposure to chemical reactions, morphological shriveling and tearing deformation may occur) might be interesting for the readers therefore it should be added in the manuscript.

Comments on the Quality of English Language

The text is readable but the English can still be improved. The authors might be satisfied with "readable". It is up to them to decide if to further improve it or not.

Author Response

Dear reviewer,

With additional modifications, our manuscript has developed considerably.

Please check the revised manuscript with interest.

Thank you again.

I hope you and your institute will continue to prosper.

Warm regards,

Gun Gyun Kim

Reviewer 4 Report

Comments and Suggestions for Authors

As a revised manuscript, the authors sincerely replied to all reviewers’ comments one-by-one.

Author Response

Dear reviewer,

Thank you for reviewing the manuscript with interest.

I hope you and your institute will continue to prosper.

Thank you again.

Warm regards,
Gun Gyun Kim